# Awareness of obstetric fistula and its associated factors among reproductive-aged women: Demographic and health survey data from Gambia

**Rabbi Tweneboah** [1]*, **Eugene Budu**[2], **Patience Dzigbordi Asiam**[3‡],
**Stephen Aguadze**[4‡], **Franklin Acheampong**[2‡]

1 Department of Economics, Kwame Nkrumah University of Science and Technology, Kumasi, Ghana,
2 Research Unit, Korle Bu Teaching Hospital, Accra, Ghana, 3 Department of Statistics, Kwame Nkrumah
University of Science and Technology, Kumasi, Ghana, 4 Korle Bu Teaching Hospital, Accra, Ghana

☯ These authors contributed equally to this work.
‡ PDA, SA and FA also contributed equally to this work.
* antwirabbi56@gmail.com

**Data Availability Statement:** The data underlying the results presented in the study are available from Measuredhs: Gambia Bureau of Statistics

## Abstract

Childbirth complications continue to remain a major problem in various settings but most rampant in underdeveloped nations, including Gambia, where poor living condition is widespread. Obstetric Fistula (OF) has been cited as one of the most common issues experienced by mothers during labor over the years. The study thus focuses on evaluating the level of awareness of this condition among Gambian women of childbearing age. Women's Data from the recent Demographic and Health Survey (DHS) in Gambia was used for the study. A total of 11,864 women of reproductive age, who had completed cases of the variables of interest were used for the analysis. Stata version-16 was used in carrying out the analysis of this study; and Pearson Chi-square test for independence was used to examine the distribution of the awareness of fistula among Gambian women across the explanatory variables. A two model binary logistic regression was fitted to examine the association between the outcome variable and the explanatory variables. The study presented that, majority of the Gambian women (87.2%) had no knowledge about Obstetric Fistula, as they indicated to have never heard of the condition. Considering the individual factors, age was seen to be a significant factor in determining the awareness level of Obstetric Fistula among women of childbearing age. As they age, the higher their odds of knowing about the condition. Other factors such as level of education, marital status, pregnancy termination, media exposure, community poverty level, and employment were also discovered to be significant factors in determining a woman's awareness of Obstetric Fistula. Considering the low level of awareness of Obstetric Fistula among Gambian women, there is therefore the need for the appropriate institutions to increase health educational programmes targeted at creating its awareness, and to provide further in-depth understanding of the condition to the few who already have a fair knowledge about it.

(GBoS)and ICF. 2021. The Gambia Demographic and Health Survey 2019-20 [Dataset]. GMIR81FL. DTA. Banjul, The Gambia and Rockville, Maryland, USA: GBoS and ICF [Producers]. ICF [Distributors], 2019-20. https://dhsprogram.com/data/dataset/Gambia_Standard-DHS_2019.cfm?flag=1

**Funding:** The authors received no specific funding for this work

**Competing interests:** The authors have declared no competing interest exist.

# Introduction

Obstetric fistula is one of the most devastating and catastrophic childbirth injuries. It is characterized by a hole which occurs between the birth canal and the bladder, where the rectum is sometimes included as well. This is caused by delayed, complicated labour without access to early, high-quality medical attention. It causes women and girls to leak fluids (i.e. faeces, urine) or both, and frequently leads to persistent medical issues, hopelessness, social isolation, and hardship [1]. Fistula affects an estimated half a million women and girls in Sub-Saharan Africa, Asia, the Arab States area, and Latin America and the Caribbean, with new instances emerging every year. Yet fistula is pretty much fully controllable [1, 2].

Its recurrence is a clear indication of vast discrepancies, a symptom of global inequality, and proof that health and wellbeing institutions are failing to safeguard the administration of human rights of the economically marginalized women and girls. Precise obstetric fistula prevalence numbers (globally and nationally) are unknown owing to unreliable database, underreporting, and embarrassment, which prevents women from making complaints about fistula. However, an estimated 50,000 to 100,000 women suffer from fistula each year, with around 2 million women already living with the condition; which is a burden in nearly 60 nations [3]. Based on data from two African countries, including The Gambia, the most recent community-based prevalence estimate is 160 (95 percent CI 116–210) obstetric fistulas per 1000 women of reproductive age [4].There are several nations in South Asia, notably Bangladesh, and in sub-Saharan Africa, such as Sudan, Ethiopia, Chad, Ghana, and Nigeria, where fistula incidence is estimated to be much prominent [5].

Several studies have postulated women's educational level, age, history of pregnancy, distance to the nearest health facility, and awareness of obstetrics problems as the primary correlates of women's awareness of obstetrics fistula [3, 6]. While prolonged labour and a lack of timely accessibility to emergency obstetric treatment are the most common leading antecedents of obstetric fistula in underdeveloped countries, widespread poverty is frequently a fundamental factor. According to studies, fistula sufferers tend to reside in rural places and are in more disadvantaged groups [7, 8].

Given the negative consequences of fistula on the health of women and girls, management and rehabilitation are major public health problems. It is projected that if all affected women received treatment, given the current pace of surgical therapy, it would take nearly 55 years to address all existing cases, not to mention the new cases that arise each year [9]. Surgical intervention to correct fistula is followed by therapy, which involves extending and moving limbs that have ceased to function as a result of genital tract and sciatica nerve injury. Lower limb paralysis, foot drop, and limb contracture require physiotherapy to be treated [10]. Psychological and emotional therapy, skill improvement, and outreach to identify women with perforations and transfer them to distant treatment clinics help solidify treatment activities [11]. Despite the high success rate of fistula treatment, up to 90%, a significant number of women remain unaware of the availability of treatment for their condition [5, 12]. Moreover, a substantial number of women do not even know that they are suffering from obstetric fistula. In response, a 2017–2021 project was launched with the goal of funding at least 150 fistula treatment procedures in The Gambia [13]. As of 2021, Gambia, in collaboration with other institutions, were able to support 19 treatment procedures, giving hope to women suffering from this illness. It is undeniable that the first step in resolving any health issue is being aware of its existence and identifying it. However, there is a paucity of evidence on the awareness of women in Gambia on OF. According to Kasamba et al. [14], limited awareness of obstetric fistula within

communities could impede affected women from seeking necessary care. In addition, several studies [15, 16] have shown that misconceptions and negative beliefs held by communities regarding fistula can discourage women from seeking medical assistance. Lyimo and Idda [17] have also suggested that greater awareness among women regarding fistula, including its risk factors, symptoms, and prevention strategies, can facilitate early detection and prompt treatment-seeking. As such, this study aims to investigate the level of obstetric fistula awareness among Gambian women of reproductive age and identify associated factors. The findings of this study will be useful in raising awareness of obstetric fistula among communities, addressing stigmatization of fistula patients, and promoting primary prevention strategies through education and sensitization efforts.

## Methods

### Data source

Data from the recent Demographic and Health Survey (DHS) in Gambia was used for the study. Specifically, the study used the women's recode file also known as the Individual Recode (IR). DHS is a comparable nationally representative survey undertaken regularly in over 90 countries, enhancing global understanding of developing country health and demographic trends [18]. The DHS Program's major goal is to improve demographic, health, and nutrition data collection, analysis, and distribution, as well as to make these data more useful for planning, policymaking, and program management [18].

### Study design and sampling procedure

A descriptive cross-sectional design was employed for the survey. Validated and pretested structured questionnaires were used to collect data from the respondents on health and social issues such as maternal health service utilization and women empowerment and sociodemographic characteristics [18, 19]. The survey was conducted using a two-stage cluster sampling technique. To begin, a stratified sample of enumeration areas (EAs) was chosen using probability proportional to size (PPS): a sample of a preset number of EAs is chosen independently in each stratum using probability proportional to the EA's measure of size. A listing technique is used in the designated EAs to ensure that all dwellings/households are listed. Second, households in the selected EAs are selected using equal probability systematic sampling. A detailed procedure for sampling has been described elsewhere [20]. We included 11,864 women with complete cases of variables of interest in the study. We also adopted the STROBE (Strengthening the Reporting of Observational Studies in Epidemiology) guidelines in drafting this manuscript [21]. The dataset is freely available to download at https://dhsprogram.com/methodology/survey/survey-display-555.cfm.

### Variables

**Outcome variable.** The outcome variable of this study was women's awareness of obstetric fistula. The variable measures the extent to which women are aware of obstetric fistula. This variable was derived from the question "have you heard about fistula?" Responses to this question were categorised into "no" and "yes". The variable was dichotomised into 1 = "ever heard of fistula" and 0 = "never heard of fistula". Studies that used the DHS dataset employed similar coding [2, 14, 22].

**Explanatory variables.** The explanatory variables considered in this study were selected based on their association with awareness of fistula from literature [2, 14, 22] and also their availability in the DHS dataset. A total of seventeen (17) variables were included in the study.

These variables can be grouped as individual and contextual factors. The individual factors considered were: Mother's age, educational level, marital status, religion, employment status, parity, wealth index, frequency of reading newspaper, frequency of listening to radio, frequency of watching television, sexual activity, pregnancy status and pregnancy termination. The contextual factors included were, type of place of residence, region, community literacy level and community poverty level. The categories of each of the variables are shown in Table 2.

## Statistical analyses

Stata version 16.0 was used to carry out the analysis in four steps. At the first stage, a graphical chart was used to summarize the results of the proportion of awareness of fistula among women in Gambia. The Pearson chi-square test of independence was adopted to examine the distribution of the awareness of fistula among women across the explanatory variables. A multi-collinearity test using the variance inflation factor (VIF) was conducted to examine the collinearity among the variables. The results indicated that the minimum, maximum, and mean VIFs were 1.04, 6.70, and 2.26 respectively; hence, there was no evidence of collinearity among the variables included in the regression analysis. Finally, a two model binary logistic regression was fitted to examine the association between the outcome variable and the explanatory variables. In the first model (Model I), there was a bivariate binary logistic regression where each of the independent variables was fitted. In the second model, Model II, which is the complete model, a multivariate binary logistic regression was fitted. Odds Ratio of 95 percent confidence intervals (95% CIs) was used to present the findings of the regression analysis. To account for disproportionate sampling and non-response, the "svyset" command was used, and weighting was done to account for the intricate nature of DHS data.

## Ethical approval and consent to participation

The survey reported that ethical approval was granted by the Institutional Review Board of ICF International and Ethical Review Committee of Gambia Health Service [23]. We further obtained permission from the DHS Program for the usage of this data for the study. The data can be accessed from their website (www.measuredhs.com). Since the DHS data is publicly available, there was no need for further ethical approval and consent to participation. More information regarding the DHS data usage and ethical guidelines can be found at http://goo.gl/ny8T6X. All methods were performed in accordance with the relevant guidelines and regulations.

## Results

### Prevalence of fistula awareness among women in Gambia

Fig 1 provides a graphical representation of the level of awareness of obstetric fistula among Gambian women considered for the study. Eighty-seven percent (87.2%) forming the majority of women in Gambia were not aware of fistula. Only 12.8% of women considered for the study had knowledge on fistula.

**Distribution of fistula awareness among women in Gambia.** Table 1 summarizes the proportion of fistula awareness in Gambia based on the socio demographic characteristics of the study. Women aged 45–49 had the highest proportion of fistula awareness (20.1%), while women aged 15–19 had the lowest proportion (6.1%). Fistula awareness among women who had never married was only (8.5%) whereas the awareness level among married women was 14.9%. Women who had four or more birth had the greatest proportion of fistula awareness (16.1%) while women with no births formed the least proportion (9.5%). The awareness level

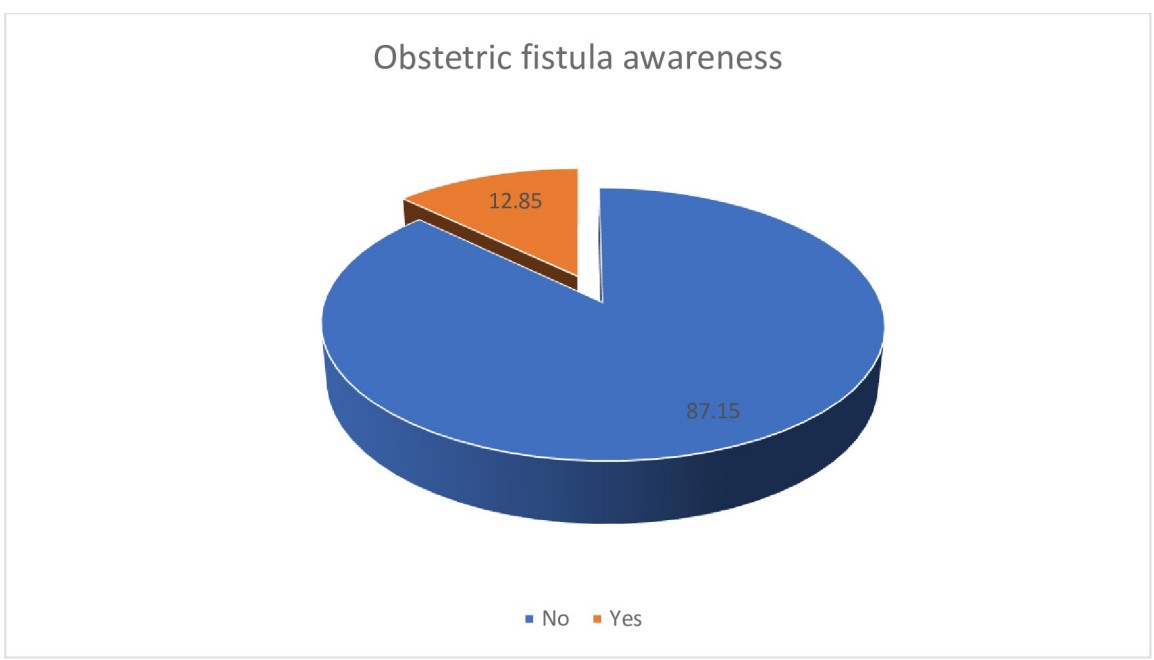

**Fig 1. Prevalence of fistula awareness among women in Gambia.**

among working women was 15.5% and that of women who are not working was 8.9%. Islamic women had the least proportion of fistula (12.6%) whilst the greatest proportion was recorded among women with other religious affiliations (41.3%).

Women who read newspaper, listened to radio and watched TV for at least once a week had the highest proportion of fistula awareness (20.2%, 14.9% and 14%) respectively. About two in ten (17.1%) women with the highest wealth index were aware of fistula. Highly educated women had the highest proportion of fistula awareness (28.5%). Only 8.4% of women who had never had sex were aware of fistula whilst women who have had sex before reported 14.6% fistula awareness. Fewer women (12.7%) who are not pregnant were aware of fistula while 14.5% pregnant women were aware of fistula. Also, women who had terminated pregnancy before had a higher fistula awareness (17.6%) whereas women who had never terminated pregnancy before had a lower fistula awareness (11.9%).

In terms of sub region, women in Juntaur recorded the highest proportion of fistula awareness (16.6%) whereas women in Janjanbureh had the least fistula awareness (7.7%). Regarding community literacy and poverty level, women in high literacy and high poverty communities had the greatest fistula awareness (14.3% and 15%) respectively. The chi-square test analysis indicated statistically substantial association between all the explanatory variables and fistula awareness except currently pregnant and type of place of residence (see, Table 1).

## Factors associated with women's awareness of obstetric fistula

The outcome of the Binary Logistic Regression has been presented in Table 2. At the bivariate level (Model 1), Compared to women in 15–19 age bracket, the highest odds of fistula awareness were women in their 45 to 49 years (AOR = 3.94, CI = 3.11–4.99), followed by 30–34 aged women (AOR = 2.92, CI = 2.38–3.58). Women aged 20–24 had less odds of fistula awareness compared to those aged 15–19. After adjusting for the covariates in model II, women aged 45–49 years were twice more likely to be aware of fistula than women aged 15–19 (COR = 2.19,

**Table 1. Distribution of fistula awareness among women in Gambia (N = 11864).**

| Variables | Weighted N | Weighted % | Fistula awareness | P-value |
|---|---|---|---|---|
| **Age** | | | | <0.001 |
| 15–19 | 2632 | 22.2 | 6.1 | |
| 20–24 | 2181 | 18.4 | 10.7 | |
| 25–29 | 2248 | 18.9 | 14.1 | |
| 30–34 | 1619 | 13.6 | 17.2 | |
| 35–39 | 1437 | 12.1 | 16.3 | |
| 40–44 | 1028 | 8.7 | 15.6 | |
| 45–49 | 718 | 6.1 | 20.1 | |
| **Marital status** | | | | <0.001 |
| Not married | 3704 | 31.2 | 8.5 | |
| Married | 7500 | 63.2 | 14.9 | |
| Cohabiting | 25 | 0.2 | 12.9 | |
| Widowed | 182 | 1.5 | 14.3 | |
| Divorced | 453 | 3.8 | 13.7 | |
| **Parity** | | | | <0.001 |
| No birth | 4321 | 36.4 | 9.5 | |
| One birth | 1457 | 12.3 | 11.4 | |
| Two births | 1239 | 10.5 | 14.4 | |
| Three births | 1205 | 10.1 | 15.4 | |
| Four or more births | 3641 | 30.7 | 16.1 | |
| **Employment status** | | | | <0.001 |
| Not working | 4752 | 40.0 | 8.9 | |
| Working | 7112 | 60.0 | 15.5 | |
| **Religion** | | | | 0.005 |
| Islam | 11442 | 96.4 | 12.6 | |
| Christianity | 418 | 3.5 | 18.6 | |
| Other | 4 | 0.1 | 41.3 | |
| **Frequency of reading newspaper** | | | | <0.001 |
| Not at all | 10124 | 85.3 | 12.0 | |
| Less than once a week | 1311 | 11.1 | 16.7 | |
| At least once a week | 429 | 3.6 | 20.2 | |
| **Frequency of listening radio** | | | | <0.001 |
| Not at all | 2693 | 22.7 | 10.8 | |
| Less than once a week | 4692 | 39.5 | 12.1 | |
| At least once a week | 4479 | 37.8 | 14.9 | |
| **Frequency of watching TV** | | | | 0.047 |
| Not at all | 2112 | 17.8 | 11.5 | |
| Less than once a week | 3144 | 26.5 | 11.4 | |
| At least once a week | 6608 | 55.7 | 14.0 | |
| **Wealth index** | | | | <0.001 |
| Poorest | 1998 | 16.9 | 11.5 | |
| Poorer | 2135 | 18.0 | 11.0 | |
| Middle | 2292 | 19.3 | 11.6 | |
| Richer | 2591 | 21.8 | 11.8 | |
| Richest | 2848 | 24.0 | 17.1 | |
| **Level of education** | | | | <0.001 |
| No education | 4119 | 34.7 | 11.8 | |

*(Continued)*

**Table 1.** (Continued)

| Variables | Weighted N | Weighted % | Fistula awareness | P-value |
|---|---|---|---|---|
| Primary | 1854 | 15.6 | 12.2 | |
| Secondary | 5020 | 42.3 | 11.2 | |
| Higher | 871 | 7.4 | 28.5 | |
| **Sexual activity** | | | | <0.001 |
| Never had sex | 3397 | 28.6 | 8.4 | |
| Ever had sex | 8467 | 71.4 | 14.6 | |
| **Currently pregnant** | | | | 0.873 |
| Not pregnant | 10984 | 92.6 | 12.7 | |
| Pregnant | 880 | 7.4 | 14.5 | |
| **Ever terminated a pregnancy** | | | | <0.001 |
| No | 9876 | 83.2 | 11.9 | |
| Yes | 1988 | 16.8 | 17.6 | |
| **Type of place of residence** | | | | 0.775 |
| Urban | 8746 | 73.7 | 13.1 | |
| Rural | 3118 | 26.3 | 12.3 | |
| **Region** | | | | <0.001 |
| Banjul | 162 | 1.4 | 14.8 | |
| Kanifing | 2589 | 21.8 | 14.5 | |
| Brikama | 5299 | 44.7 | 12.9 | |
| Mansakonko | 431 | 3.6 | 15.7 | |
| Kerewan | 1128 | 9.5 | 10.6 | |
| Juntaur | 523 | 4.4 | 16.6 | |
| Janjanbureh | 595 | 5.0 | 7.7 | |
| Basse | 1136 | 9.6 | 10.8 | |
| **Community literacy level** | | | | 0.001 |
| Low | 2163 | 18.2 | 10.3 | |
| Medium | 4271 | 36.0 | 12.3 | |
| High | 5430 | 45.8 | 14.3 | |
| **Community poverty level** | | | | <0.001 |
| Low | 4379 | 36.9 | 11.7 | |
| Moderate | 1833 | 15.4 | 9.1 | |
| High | 5652 | 47.6 | 15.0 | |

CI = 1.55–3.10). Also, women aged 40–44 had higher odds of fistula awareness compared to women aged 15–19 (COR = 1.71, CI = 1.23–2.38).

With married women as the reference, unmarried were 46% less likely to be aware of fistula (AOR = 0.54, CI = 0.47–0.62). Similar results were obtained after adjusting for the other socio demographic factors, unmarried women had less odds of fistula awareness (COR = 0.70, CI = 0.53–0.92). Compared to women with no birth, women with four or more births had the highest odds of fistula awareness (AOR = 1.92, CI = 1.68–2.20), followed by women with three births (AOR = 1.69, CI = 0.79–1.39). Women with one birth are 1.28 times more likely to be aware of fistula than women with no birth (AOR = 1.28, CI = 1.06–1.55); then women that are working are more likely to be aware of fistula than women that are not working (AOR = 1.88, CI = 1.66–2.12). Again, Christian women were 1.60 times more likely to be aware of fistula than Islamic women.

**Table 2. Association between explanatory variables and fistula awareness among women in Gambia.**

| Variable | Model I | Model II |
|---|---|---|
| | AOR (95% CI) | COR (95% CI) |
| **Age** | | |
| 15–19 | Reference (1.0) | Reference (1.0) |
| 20–24 | 1.81[***] (1.47–2.23) | 1.35[*] (1.07–1.70) |
| 25–29 | 2.38[***] (1.96–2.90) | 1.55[***] (1.20–2.02) |
| 30–34 | 2.92[***] (2.38–3.58) | 1.70[***] (1.26–2.28) |
| 35–39 | 2.88[***] (2.34–3.55) | 1.68[**] (1.23–2.29) |
| 40–44 | 2.89[***] (2.30–3.62) | 1.71[**] (1.23–2.38) |
| 45–49 | 3.94[***] (3.11–4.99) | 2.19[***] (1.55–3.10) |
| **Marital status** | | |
| Not married | 0.54[***] (0.47–0.62) | 0.70[*] (0.53–0.92) |
| Married | Reference (1.0) | Reference (1.0) |
| Cohabiting | 0.84 (0.19–3.68) | 0.65 (0.15–2.84) |
| Widowed | 0.99 (0.62–1.53) | 0.83 (0.53–1.32) |
| Divorced | 1.11 (0.84–1.46) | 0.86 (0.64–1.17) |
| **Parity** | | |
| No birth | Reference (1.0) | Reference (1.0) |
| One birth | 1.28[*] (1.06–1.55) | 0.96 (0.73–1.24) |
| Two births | 1.40[***] (1.15–1.71) | 0.91 (0.70–1.20) |
| Three births | 1.69[***] (1.40–2.05) | 1.05 (0.79–1.39) |
| Four or more births | 1.92[***] (1.68–2.20) | 1.24 (0.94–1.61) |
| **Employment status** | | |
| Not working | Reference (1.0) | Reference (1.0) |
| Working | 1.88[***] (1.66–2.12) | 1.43[***] (1.25–1.64) |
| **Religion** | | |
| Islam | Reference (1.0) | Reference (1.0) |
| Christianity | 1.60[***] (1.18–2.17) | 1.22 (0.86–1.71) |
| Other | 3.40 (0.31–37.55) | 1.06 (0.89–12.74) |
| **Frequency of reading newspaper** | | |
| Not at all | Reference (1.0) | Reference (1.0) |
| Less than once a week | 1.68[***] (1.42–1.99) | 1.30[**] (1.06–1.58) |
| At least once a week | 2.15[***] (1.64–2.83) | 1.13 (0.82–1.56) |
| **Frequency of listening radio** | | |
| Not at all | Reference (1.0) | Reference (1.0) |
| Less than once a week | 1.25[***] (1.07–1.44) | 1.16 (0.99–1.36) |
| At least once a week | 1.42[***] (1.23–1.64) | 1.19[*] (1.02–1.39) |
| **Frequency of watching TV** | | |
| Not at all | 0.93 (0.81–1.06) | 1.04 (0.87–1.23) |
| Less than once a week | 0.85[*] (0.74–0.97) | 0.97 (0.84–1.12) |
| At least once a week | Reference (1.0) | Reference (1.0) |
| **Wealth index** | | |
| Poorest | Reference (1.0) | Reference (1.0) |
| Poorer | 0.85 (0.72–1.01) | 0.93 (0.77–1.12) |
| Middle | 0.99 (0.84–1.16) | 1.05 (0.861.29) |
| Richer | 0.95 (0.81–1.13) | 0.98 (0.76–1.26) |
| Richest | 1.46[***] (1.25–1.70) | 1.22 (0.93–1.61) |
| **Level of education** | | |

(*Continued*)

**Table 2.** (Continued)

| Variable | Model I | Model II |
|---|---|---|
| | AOR (95% CI) | COR (95% CI) |
| No education | Reference (1.0) | Reference (1.0) |
| Primary | 1.09 (0.93–1.28) | 1.35*** (1.14–1.59) |
| Secondary | 0.95 (0.84–1.08) | 1.38*** (1.17–1.62) |
| Higher | 3.27*** (2.70–3.96) | 3.54*** (2.74–4.59) |
| **Sexual activity** | | |
| Never had sex | 0.53*** (0.46–0.61) | 1.03 (0.74–1.43) |
| Ever had sex | Reference (1.0) | Reference (1.0) |
| **Ever terminated a pregnancy** | | |
| No | Reference (1.0) | Reference (1.0) |
| Yes | 1.58*** (1.39–1.79) | 1.21*** (1.06–1.40) |
| **Region** | | |
| Banjul | Reference (1.0) | Reference (1.0) |
| Kanifing | 0.93 (0.74–1.17) | 0.96 (0.75–1.21) |
| Brikama | 0.83 (0.68–1.04) | 0.95 (0.75–1.20) |
| Mansakonko | 1.12 (0.88–1.432) | 1.60** (1.17–2.18) |
| Kerewan | 0.65*** (0.51–0.83) | 0.95 (0.70–1.30) |
| Juntaur | 1.12 (0.89–1.41) | 1.92*** (1.39–2.65) |
| Janjanbureh | 0.55*** (0.42–0.72) | 0.85 (0.60–1.20) |
| Basse | 0.69** (0.55–0.87) | 1.13 (0.83–1.54) |
| **Community literacy level** | | |
| Low | Reference (1.0) | Reference (1.0) |
| Medium | 1.05 (1.10–1.43) | 1.13 (0.95–1.35) |
| High | 1.26*** (1.10–1.43) | 1.12 (0.88–1.44) |
| **Community poverty level** | | |
| Low | Reference (1.0) | Reference (1.0) |
| Moderate | 0.88 (0.74–1.05) | 0.90 (0.73–1.12) |
| High | 1.29*** (1.15–1.45) | 1.17 (0.91–1.51) |

{***} {**} {*} shows 1%, 5% and 10% significance respectively

AOR: Adjusted odds ratio

COR: Crude odds ratio

Model I: Bivariate analysis between fistula awareness and each explanatory variable

Model II: Multivariate analysis between fistula awareness and the explanatory variables

Women that listen to radio and read newspapers at least once a week had higher odds of fistula awareness than women that do not listen to radio or read newspapers (respectively; AOR = 1.40, CI = 1.23–2.83 & AOR = 2.15, CI = 1.64–2.83). After adjusting for the covariates, women who listen to radio at least once a week and read newspapers less than once a week had a higher odds of fistula awareness than women who do not listen to radio or read newspapers. Compared to the poorest women in Gambia, women with the wealthiest background were 1.46 times more likely to be aware of fistula (COR = 1.46, CI = 1.25–1.70). Women with higher education are approximately 3 times more likely to be aware of fistula than women with no education (AOR = 3.27, CI = 2.70–3.96). Adjusting for the covariate effects, women with higher education had the highest odds of fistula awareness (COR = 3.54, CI = 2.74–4.59), followed by women with secondary education (COR = 1.38, CI = 1.17–1.62), compared to women with no education. Also, women who have never had sex were 47% less likely to be

aware of fistula than women who have had sex. Women who have had abortion were more likely to be aware of fistula, compared to women who had never terminated pregnancy. Regarding sub region, women from Janjanbureh had the least odds of fistula awareness (AOR = 0.55, CI = 0.42–0.72), followed by women in Kerewan (AOR = 0.65, CI = 0.51–0.83), then by Basse women (AOR = 0.69, CI = 0.55–0.87), compared to women in Banjul region. Adjusting for the effects of other factors in model II, women in Juntaur region had 1.92 (CI = 1.39–2.65) and those in Mansakonko had 1.60 (CI = 1.17–2.18). With women from low literacy community and women from low poverty community as the reference, women from high literacy community had greater odds of fistula awareness (AOR = 1.26, CI = 1.10–1.43) and women from high poverty community had 1.269 (1.15–1.45).

## Discussion

Mitigating obstetric fistula (OF) is critical to the attainment of SDG target 3.1. The study looked at the level of awareness obstetric fistula and the factors that influence it in Gambian women of reproductive age. Overall, the prevalence of OBF awareness in Gambia was 12.9%. This result was much lower than studies in Nigeria and Tanzania (57.8% and 60.1%) [24, 25] respectively. A plausible justification for this finding can be attributed to the differences in study period, design, and sample size. Nevertheless, the finding suggest that fistula awareness is very low in Gambia. This may have a detrimental effect on African countries' quest to mitigate obstetric fistula as women in Gambia are more likely to stay home rather than seek medical intervention [26].

According to the findings, age, marital status, employment position, media exposure, and educational level were all significant socio-demographic factors associated with fistula awareness. The likelihood of being aware of fistula increases with age (e.g. women in 45–49 age bracket were twice likely to be aware of fistula compared to those in 15–19). That is, older women are more likely to be aware of fistula than younger women. This result is in line with prior research from Ethiopia [27] and Uganda [28]. The fundamental assumption behind this assertion is that, the older a woman gets, the more experience she would have with birth and its concomitant complications. Our study also revealed that unmarried women in Gambia are less likely to be aware of fistula compared to married women. This is largely due to the influence of the husbands of the married women. Married women, due to the influence of their partners are more likely to seek healthcare services such as obstetric counselling which creates an avenue of increasing awareness among them. There is also the possibility of knowledge transfer from husband to wife. The decision making power of a Gambian woman to visit health institution during labour is very low, the authority is largely made by their husbands [29].

Compared to women who were not exposed to mass media (i.e. reading newspapers and listening to radio), women who were exposed to mass media were more likely to be aware of obstetric fistula. The findings of prior investigations in Nigeria [22] and Ethiopia [30] complement the findings of this study. The finding is probably due to the fact that the media is a vital conduit for conveying information, such as information regarding OF, its signs and symptoms, and information about where to get treatment. As enshrined in [6], most women who are aware of fistula got the information from the media.

Higher level of education among women of reproductive age directly translate to higher health literacy. Education level is a significant predictor of women's awareness of fistula in Gambia. It has been established in several studies [31] that education gradient positively associate with better health behaviours (such as seeking obstetric counselling) and improved health status (fistula awareness). Several other studies [6, 32, 33] had similar findings that women

with higher education are more likely to have improved health status (aware of fistula) compared to those without formal education. Formal education empowers women to make better healthcare decision such as attending health education forums and alter habits that are detrimental to ones' health.

As expected, there is a strong association between fistula awareness and pregnancy termination. Women who had history or were exposed to pregnancy termination were 1.21 times more likely to have better awareness of fistula than those who had not. The finding could be rationalized by the fact that pursuing healthcare services such as pregnancy termination services presents an avenue for women to be exposed to health education and promoting messages, perhaps raising their awareness of fistula. Similar finding was reported by Aleminew et al. [26] and Asefa et al. [30].

Compared to Banjul, which is a major urban concentration in Gambia, women in these rural towns (Juntaur and Mansakonko) are 1.92 and 1.60 times more likely to be aware of fistula. The high prevalence of obstetric fistula in rural areas in Gambia [34] could explain why awareness of fistula is high also in these towns. However, the finding deviates from the findings of several similar studies which posits that awareness is rather high in urban areas [6, 29].

## Strength and weakness

By far, our literature review shows that this is the first study to use nationally representative datasets from Gambia to investigate the prevalence of and factors associated with obstetric fistula awareness. This is a significant addition to the existing literature. However, there are some downsides to consider. The DHS does not break down the question to assess which type of fistula women are aware of (e.g., vesicovaginal fistula, urethrovaginal fistula, or rectovaginal fistula). In the future, this disaggregation could be considered for the DHS dataset. Additionally, the inclusion of health workers in the occupational classification of DHS data would enable future studies to assess the awareness levels of health workers regarding obstetric fistula. Also, the cross-sectional study design does not allow us draw direct causal inference to the factors associated with obstetric fistula awareness.

## Conclusion

The present study sought to assess the magnitude and factors associated with fistula awareness in Gambia. We conclude that obstetric fistula awareness is very low in Gambia. As such, immediate remediating action is needed to raise women's awareness about OF. It is evident from the study that age, level of education, marital status, pregnancy termination, media exposure, community poverty level, and employment status were significant factors associated with OF awareness. To raise women's awareness of fistula, there is the need for public health interventions to consciously raise community literacy rate, increase access to mass media platforms and invest intensively in formal education for women.

## Supporting information

**S1 File. Contains all supporting information regarding the data used for the study.**
(ZIP)

## Acknowledgments

We thank measuredhs for giving us access to the dataset.

## Author Contributions

**Conceptualization:** Rabbi Tweneboah, Eugene Budu.

**Data curation:** Rabbi Tweneboah, Eugene Budu.

**Formal analysis:** Rabbi Tweneboah, Eugene Budu.

**Methodology:** Rabbi Tweneboah, Eugene Budu.

**Supervision:** Franklin Acheampong.

**Writing – original draft:** Rabbi Tweneboah, Patience Dzigbordi Asiam.

**Writing – review & editing:** Rabbi Tweneboah, Eugene Budu, Patience Dzigbordi Asiam, Stephen Aguadze, Franklin Acheampong.

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
