## [Decision Letter · Decision Letter 0]

8 Feb 2023

PONE-D-22-16318AWARENESS OF OBSTETRIC FISTULA AND ITS ASSOCIATED FACTORS AMONG REPRODUCTIVE-AGED WOMEN: DEMOGRAPHIC AND HEALTH SURVEY DATA FROM GAMBIAPLOS ONE

Dear Dr. Rabbi Tweneboah,

Thank you for submitting your manuscript to PLOS ONE. After careful consideration, we feel that it has merit but does not fully meet PLOS ONE’s publication criteria as it currently stands. Therefore, we invite you to submit a revised version of the manuscript that addresses the points raised during the review process.

We look forward to receiving your revised manuscript.

Kind regards,

Sidrah Nausheen, FCPS

Academic Editor

PLOS ONE

Journal Requirements:

 Whilst you may use any professional scientific editing service of your choice, PLOS has partnered with both American Journal Experts (AJE) and Editage to provide discounted services to PLOS authors. Both organizations have experience helping authors meet PLOS guidelines and can provide language editing, translation, manuscript formatting, and figure formatting to ensure your manuscript meets our submission guidelines. To take advantage of our partnership with AJE, visit the AJE website (http://aje.com/go/plos) for a 15% discount off AJE services. To take advantage of our partnership with Editage, visit the Editage website (www.editage.com) and enter referral code PLOSEDIT for a 15% discount off Editage services. If the PLOS editorial team finds any language issues in text that either AJE or Editage has edited, the service provider will re-edit the text for free.

 ● A clean copy of the edited manuscript (uploaded as the new *manuscript* file).

Reviewers' comments:

Reviewer's Responses to Questions

**Comments to the Author**

1. Is the manuscript technically sound, and do the data support the conclusions?

Reviewer #1: Yes

Reviewer #2: Yes

2. Has the statistical analysis been performed appropriately and rigorously? 

Reviewer #1: I Don't Know

Reviewer #2: Yes

3. Have the authors made all data underlying the findings in their manuscript fully available?

Reviewer #1: Yes

Reviewer #2: Yes

4. Is the manuscript presented in an intelligible fashion and written in standard English?

Reviewer #1: Yes

Reviewer #2: Yes

5. Review Comments to the Author

Reviewer #1: The research in this manuscript is done on the wrong population.It is obvious that women do not have knowledge of fistula as its an obstetrical complication.The research question of awareness should have been posed to the care giver (lady health visitor, midwife,obstetrician).

The purpose statement is not complete what benefit would it be for doing this research on the community?

If the awareness of women is required than they support it with evidence as to why? and how the awareness of women would decrease the incidence in this population

Reviewer #2: the paper is well written and with all the required details in statistical analysis.

The manuscript is a technically sound piece of scientific research with data that supports the conclusions. it has been conducted adequately with appropriate sample size. The conclusions have been drawn appropriately based on the data presented in standard english.

6. PLOS authors have the option to publish the peer review history of their article (what does this mean?). If published, this will include your full peer review and any attached files.

Reviewer #1: No

Reviewer #2: No

---

## [Author Response · Author response to Decision Letter 0]

10 Mar 2023

Reviewer 1

General comments:

The research in this manuscript is done on the wrong population. It is obvious that women do not have knowledge of obstetric fistula. The purpose statement is not complete.

Response: 

Thank you for your constructive review, the reviewer’s comments have improved our manuscript.

Specific comments:

Comment 1. The research is done on the wrong population. It is obvious women do not have knowledge of fistula because it is an obstetric complication. 

Response: 

Thank you for your comment. We appreciate the feedback. However, we respectively disagree with the notion that the study is done on the wrong population. We believe women are the ones that are primarily affected by obstetric fistula, and they are the ones that face the physical and emotional burden of the condition. As such, it is essential to investigate their awareness level of the condition as it is the first step in addressing it (early detection and reporting). Moreover, the focus of the study is not on the awareness of the condition among women, but rather the factors that influence their awareness level, as several studies have documented its importance to addressing obstetric fistula.

Comment 2. The research question of awareness should have been posed to the caregivers (lady health visitor, midwife, obstetrician).

Response: The authors acknowledge that awareness of obstetric fistula among health workers is important and should be subjected to future research. However, the emphasis and gap identified in the study is on the community, particularly, among women. This comment has helped us to improve the strength and weakness section of the study.

Comment 3. The purpose of the study is not complete, what benefit would it be for doing this research on the community.

Response:

Thank you. We have improve the purpose statement of the study.

Comment 4. If the awareness of women is required, then they have to provide evidence to support as to why.

Response:We agree that our initial submission did not stress on empirical evidence of the importance of women awareness of obstetric fistula. Improvements have been made regarding this. Thank you. 

Comment 5. How would women’s awareness decrease the incidence of obstetric fistula in the population?

Response: Since the emphasis of the study is on increasing the awareness of women regarding obstetric fistula and its associated risk factors, we believe it is likely to decrease the prevalence of the condition among the population. This is because, increased knowledge of fistula among women will likely lead to seeking timely and appropriate medical care. Also, improved knowledge of fistula will alter the health-related behaviours among women, thereby reducing the incidence of the condition.

Reviewer 2

General comments:

The paper is well written and with all the required details in statistical analysis.

The manuscript is a technically sound piece of scientific research with data that supports the conclusions. It has been conducted adequately with appropriate sample size. The conclusions have been drawn appropriately based on the data presented in standard English.

Response: 

We appreciate and thank you for your review and comments.

Academic Editor

General comments:

Both the reviewers and academic editor feel that the study has merit but does not fully meet PLOS ONE’s publication criteria as it currently stands.

Response: 

We appreciates the academic editor’s and reviewer’s comments; we have made a revised version of the manuscript that addresses each point raised during the review process. 

Specific comments:

Comment 1. Please ensure that the manuscript meets PLOS ONE’s style requirements. 

Response: 

The revised version of the manuscript has been restructured to meet PLOS ONE’s style requirement.

Comment 2. We suggest you thoroughly copyedit the manuscript for language usage, spelling, and grammar.

Response: 

Thank you. We made changes throughout the manuscript for language usage, spelling, and grammar.

Comment 3. Your ethics should only appear in the Methods section of your manuscript. If your ethics statement is written in another section besides the Methods, please delete it.

Response:

Thank you for your comment. The ethics statement has been deleted from other sections of the manuscript.

---

## [Editor Report · Decision Letter 1]

14 Mar 2023

AWARENESS OF OBSTETRIC FISTULA AND ITS ASSOCIATED FACTORS AMONG REPRODUCTIVE-AGED WOMEN: DEMOGRAPHIC AND HEALTH SURVEY DATA FROM GAMBIA

PONE-D-22-16318R1

Dear Dr. Rabbi Tweneboah,

We’re pleased to inform you that your manuscript has been judged scientifically suitable for publication and will be formally accepted for publication once it meets all outstanding technical requirements.

Kind regards,

Sidrah Nausheen, FCPS

Academic Editor

PLOS ONE
---

## [Editor Report · Acceptance letter]

30 Mar 2023

PONE-D-22-16318R1 

Awareness of Obstetric Fistula and its Associated Factors among Reproductive-Aged Women: Demographic and Health Survey Data from Gambia 

Dear Dr. Tweneboah:

I'm pleased to inform you that your manuscript has been deemed suitable for publication in PLOS ONE. Congratulations! Your manuscript is now with our production department. 

Kind regards, 

on behalf of

Dr. Sidrah Nausheen 

Academic Editor

PLOS ONE